# Removal Efficiency and Mechanism for Cl^−^ from Strongly Acidic Wastewater by VC-Assisted Cu_2_O: Comparison Between Synthesis Methods

**DOI:** 10.3390/toxics13100890

**Published:** 2025-10-17

**Authors:** Ying Yu, Dong Li, Jialin Ma, Zhoujing Yan, Haoran Liu, Wenyue Dou, Haotian Hao

**Affiliations:** 1School of Environmental Science and Engineering, Xiamen University of Technology, Xiamen 361024, China; yingyu_2021@163.com; 2School of Ecology and Environment, Xuzhou Vocational College of Bioengineering, Xuzhou 221006, China; 19852490716@163.com; 3Jiangsu Key Laboratory of Industrial Pollution Control and Resource Reuse, School of Environmental Engineering, Xuzhou University of Technology, Xuzhou 221018, China; linxittkx@163.com (J.M.); a15380305513@163.com (Z.Y.); 18168618023@163.com (H.L.); 4Key Laboratory of Environmental Aquatic Chemistry, State Key Laboratory of Regional Environment and Sustainability, Research Center for Eco-Environmental Sciences, Chinese Academy of Sciences, Beijing 100085, China; hthao@rcees.ac.cn

**Keywords:** chloride ion, strongly acidic wastewater, cuprous oxide, precipitation, ascorbic acid

## Abstract

The discharge of strongly acidic industrial wastewater containing high concentration of chloride ions (Cl^−^) has become one of the major environmental challenges faced globally. For the removal of extremely stable Cl^−^ in acidic aqueous conditions, precipitation method possesses major advantages of strong adaptability and simple operation. This study proposed a novel cuprous oxide (Cu_2_O) method assisted by ascorbic acid (VC) for the removal of Cl^−^ from strongly acidic wastewater. First, liquid-phase reduction was chosen as the optimal Cu_2_O synthesis method based on product purity and composition. Then, parameter optimization results show that increased reagent dosage and acidity significantly enhanced Cl^−^ removal efficiency, while other factors had negligible impacts. After treatment with the sole addition of Cu_2_O, the dosed Cu_2_O existed in four forms, including cuprous chloride (CuCl), copper ion (Cu^2+^), elemental copper (Cu^0^), and Cu_2_O, among which the generation of Cu^2+^ and Cu^0^, through the oxidation and disproportionation of cuprous ion (Cu^+^), served as the main reason for the unsatisfactory efficiency in the removal of Cl^−^. Fortunately, VC is precisely capable of inhibiting the side reactions of Cu^+^, and under the assistance of 0.10 g VC, the removal of Cl^−^ by Cu_2_O was greatly improved with the multiple of theoretical reagent dosage decreasing from 12 to 3, the residual concentration of Cu^2+^ decreasing from 1197 to 18.4 mg/L and the residual concentration of Cl^−^ decreasing from 88.4 to 53.8 mg/L, thus validating the feasibility of this method.

## 1. Introduction

With the rapid development of chemical engineering, metallurgy, and electroplating industries, the discharge of strongly acidic industrial wastewater containing high concentration of chloride ions (Cl^−^) has become one of the major environmental challenges faced globally. The composition of this kind of wastewater is closely related to specific production processes. For instance, in the steel industry, hydrochloric acid is commonly used to remove the surface oxide layer of steel during the steel surface treatment process, leading to the existence of Cl^−^ and ferrous ion (Fe^2+^) in wastewater with a pH below 2 [1,2,3]. In the acid washing process for pre-treatment of electroplating, wastewater with a pH of 1 to 2 containing heavy metals and Cl^−^ is usually produced [4,5,6]. Furthermore, desulfurization wastewater, a type of sulfuric acid (H_2_SO_4_) wastewater containing high concentrations of heavy metals, fluoride (F^−^) and Cl^−^, is generated in coal-fired power plants and other enterprises [7,8,9,10]. This type of wastewater is highly acidic and extremely corrosive to equipment; furthermore, the high concentration of Cl^−^ ranging from several to tens of thousands of mg/L far exceeds the discharge limits. Once released into the environment, Cl^−^ will exacerbate eutrophication of water bodies and salinization of soil [11]. Meanwhile, Cl^−^ can interact synergistically with the acidic environment to accelerate metal corrosion and affect the normal operation of devices [12]. In addition, the coexistence of complex pollutants including heavy metals, organic chlorides, F^−^, etc., further increases the difficulty of wastewater treatment.

As the most stable form of chlorine element, Cl^−^ in acidic aqueous conditions possesses extremely stable electronic structure and high chemical inertness, making its removal relatively difficult [13,14]. Additionally, for most organisms, Cl^−^ does not participate in energy metabolism, and thus is not utilized by microorganisms as an energy source, nor can it be degraded through metabolic pathways such as respiration or fermentation. The main removal pathways for Cl^−^ include substitution by other anions and co-removal with other cations [15,16]. Currently, precipitation, evaporation concentration, membrane separation, adsorption, electrolysis, solvent extraction, and ion exchange are common approaches for the treatment of wastewater containing Cl^−^ [15,16,17,18,19]. However, in strongly acidic conditions, many of the above-mentioned methods are no longer applicable. As an example, membrane separation technology often faces bottlenecks such as acid corrosion and severe membrane fouling in acidic wastewater [20,21]. In recent years, although new types of adsorption materials and electrochemical oxidation coupling technology have been developed specially for the removal of Cl^−^ from acidic wastewater, the dual limitations of cost and efficiency in practical applications still need to be overcome [22]. Therefore, developing efficient, economical, and environmentally friendly removal technologies for Cl^−^ has become a key issue that urgently needs to be addressed in the field of strongly acidic wastewater treatment.

Compared with other technologies, precipitation method possesses major advantages of strong adaptability and simple operation, attracting considerable attention from enterprises [15,16]. From the perspectives of theoretical mechanisms and economic feasibility, cuprous (Cu^+^) and bismuth ion (Bi^3+^) are both potential precipitants for Cl^−^ under acidic conditions. Nevertheless, the removal of Cl^−^ by Bi^3+^ still remains in the laboratory research stage and lacks industrial application cases [23,24,25]. The method of removing Cl^−^ using copper slag, whose active ingredients include zero-valent copper (Cu^0^) and divalent copper (Cu^2+^), has been applied in certain metallurgical enterprises for the treatment of desulfurization wastewater [16]. During the treatment, comproportionation first occurs between Cu^0^ and Cu^2+^ to generate Cu^+^ (Equation (1)), and next, Cl^−^ precipitates with Cu^+^ and is removed as cuprous chloride (CuCl) precipitates according to Equation (2) [16]. Nevertheless, the disadvantages of low efficiency and high precipitant dosage have led enterprises to reconsider the application value of this method [26]. In previous studies, it was found that ultraviolet (UV) helped solve the above problems to a certain extent, yet inevitably, the use of UV increased energy consumption and operational difficulty [27]. In view of this, further improvement in the removal of Cl^−^ from strongly acidic wastewater through precipitation method is necessary.(1)$${\mathrm{Cu}}^{2+}+\;{\mathrm{Cu}}^{0}\to \;{2\mathrm{Cu}}^{+}$$(2)$${\mathrm{Cu}}^{+}+{\mathrm{Cl}}^{-}\;\to \;\mathrm{CuCl}↓\;$$

The high precipitant dosage and required acidity for the removal of Cl^−^ using copper slag originate from the extremely high difficulty of the comproportionation between Cu^0^ and Cu^2+^ to generate Cu^+^. In view of this, it was hypothesized that the direct addition of reagents containing Cu^+^, instead of Cu^0^ and Cu^2+^, might significantly improve the original precipitation approach. Among various Cu^+^ compounds, cuprous oxide (Cu_2_O) seems to be the most suitable for application in the removal of Cl^−^, thanks to its relatively simple preparation process, strong stability, and avoidance of introducing secondary pollutants [28,29].

In this work, a method for the removal of Cl^−^ from strongly acidic wastewater using Cu_2_O was studied. First, synthesis methods for Cu_2_O were compared and selected based on product purity and composition. Next, the effects of temperature, reagent dosage, H_2_SO_4_ concentration, initial Cl^−^ concentration, and reaction time were systematically explored. Subsequently, the crystal structure, surface morphology, and elemental valence for the precipitates were characterized to determine the mechanism. Furthermore, ascorbic acid (VC) was employed as a co-reducing agent to improve the effect of Cu_2_O on Cl^−^ removal. Finally, Cl^−^ removal using the synthesized Cu_2_O was conducted in actual strongly acidic wastewater from different industries to explore its applicability.

## 2. Experimental Section

### 2.1. Reagents and Materials

The employed copper chloride (CuCl_2_), potassium nitrate (KNO_3_), potassium chloride (KCl), sodium chloride (NaCl), sodium hydroxide (NaOH), copper sulfate (CuSO_4_·5H_2_O), trisodium citrate (TSC, C_6_H_5_Na_3_O_7_·2H_2_O), phenolphthalein (C_20_H_14_O_4_), VC, nitric acid (HNO_3_), H_2_SO_4_, ethanol (C_2_H_6_O), polyvinylpyrrolidone (PVP), and standard solution of copper (1000 µg/mL) were all purchased from Sinopharm Chemical Reagent Co., Ltd. (Shanghai, China). Hydrogen sulfide (H_2_S) and nitrogen (N_2_) gases were obtained from Nanjing Special Gas Co., Ltd. (Nanjing, Jiangsu, China). The deionized (DI) water was produced from a Milli-Q water purification system (Millipore, Billerica, MA, USA) and was employed for the preparation of all samples.

### 2.2. Synthesis of Cu_2_O

Three types of Cu_2_O samples were prepared using deposition, liquid-phase reduction, and hydrothermal methods, respectively, named COA, COB, and COC. The detailed synthesis steps are elaborated as follows.

Synthesis steps for COA using deposition method: (1) The flask containing 300 mL of DI water and 5.13 g of CuCl_2_ was placed in the water bath under 55 °C first. (2) Next, 15 g of PVP and 100 mL of DI water containing 24 g of NaOH were successively added into the above-mentioned flask. (3) After 30 min of magnetic stirring (400 r/min) on the mixed solution, 100 mL of DI water with 31.8 g of VC was continuously added into the flask. (4) After 3 h of reaction, the solid products were separated by vacuum filtration, washed by DI water and absolute ethanol successively, and dried at 40 °C for 12 h in a drying oven (LC-101-1B, Lichen Technology Co., Ltd., Changsha, Hunan, China) for next employment.

Synthesis steps for COB using liquid-phase reduction method: (1) An amount of 40 mL of DI water with 2.4 g of NaOH was first added into the flask containing 20 mL of DI water with 2.5 g of CuSO_4_·5H_2_O. (2) After 30 min of magnetic stirring (400 r/min) on the mixed solution, 50 mL of DI water with 0.88 g of VC was continuously added into the flask. (3) After 30 min of reaction, the solid products were separated by vacuum filtration, washed by DI water and absolute ethanol successively, and dried at 40 °C for 12 h in a drying oven for next employment.

Synthesis steps for COC using hydrothermal method: (1) 10 mL of DI water with 0.32 g of NaOH was first added into the flask containing 40 mL of DI water with 1 g of CuSO_4_·5H_2_O. (2) After 30 min of magnetic stirring (400 r/min) on the mixed solution, 2 g of VC was continuously added into the flask. (3) And then, the flask was placed in the water bath under magnetic stirring at 90 °C. (4) After 10 h of reaction, the solid products were separated by vacuum filtration, washed by DI water and absolute ethanol successively, and dried at 40 °C for 12 h in a drying oven for next employment.

### 2.3. Removal of Cl^−^ Using Cu_2_O or VC-Assisted Cu_2_O

Experiments for the removal of Cl^−^ from the simulated wastewater were conducted in a 100 mL flask on a magnetic stirring apparatus (LC-UMS-6, Lichen Technology Co., Ltd., Changsha, Hunan, China) at 400 r/min under different temperatures, reagent dosages, H_2_SO_4_ concentrations, initial Cl^−^ concentrations, and reaction times. After the addition of the synthesized Cu_2_O at a certain dosage into the reactor, 2.00 mL of samples were taken at certain time intervals and filtered through a 0.45 μm cellulose acetate membrane for the determination of Cl^−^ and Cu^2+^ concentrations. Furthermore, for comparison, VC with a certain dosage was added together with Cu_2_O in the improved method. After reaction, the solid products were separated by vacuum filtration, washed with DI water and dried at 40 °C for 12 h in a drying oven for subsequent characterization.

The involved actual wastewaters include electrolytic wastewater from a chlor-alkali industry, desulfurization wastewater from a copper smelter, and metal surface polishing wastewater from a steel works, the major compositions of which are shown in Table 1. The operation for the treatment of actual wastewater was similar to that of simulated wastewater, with the difference being that the 100 mL flask was replaced with a 5 L cylindrical glass reactor equipped with a stirring paddle (100 r/min) containing 5 L of wastewater.

### 2.4. Analytical Methods

Cl^−^ in the sample was quantified through the ion selective electrode (ISE) standard addition method, the detailed steps for which can be referred to in the previous works [30,31]. The concentration of Cl^−^ can be calculated as shown in Equation (3). Additionally, Cu^2+^ was quantified by an inductively coupled plasma optical emission spectrometer (ICP-OES) (NexION 300X, PerkinElmer, Waltham, MA, USA).(3)$$[{\mathrm{Cl}}^{-}]=\frac{35.5}{10^{(E_{1}\;-{\;E}_{2})/54.89}-1}\times \;50\;\mathrm{mg}/L$$

A UV-vis spectrophotometer was employed for the determination of intermediate active species in the reaction solutions. The crystal texture, surface morphology with elemental distribution, and elemental valence of the solid products were recorded by X-ray diffraction (XRD) (X’Pert PRO MPD, PANalytical, Almelo, Overijssel, The Netherlands), scanning electron microscopy with energy dispersive X-ray spectroscopy (SEM-EDS) (SU-8020, Hitachi, Tokyo, Japan), and X-ray photoelectron spectroscopy (XPS) (ESCALAB 250Xi, Thermo Fisher, Waltham, MA, USA), respectively.

## 3. Results and Discussion

### 3.1. Removal of Cl^−^ Under Different Parameters

To determine the purity and composition of the synthesized Cu_2_O, XRD and SEM analyses were first performed, and the results are shown in Figure 1. According to Figure 1a, the sharp peaks for COA and COB at 2-Theta of 29.6°, 36.4°, 42.3°, and 61.4° signify rather high crystallinity. In addition, no obvious signals of other substances were obtained, indicating that both COA and COB exhibited high purity. However, strong signals of Cu^0^ at 2-Theta of 43.1° and 50.2° were observed for COC, implying that in the hydrothermal method employed, Cu^2+^ was easily reduced to Cu^0^ rather than Cu^+^. From the morphology in Figure 1b, COA and COB, respectively, exhibited polyhedral and cubic shapes. Although the sizes of COA and COB are both around 1 µm, the size distribution of COB appears more uniform. Moreover, the aggregation degree of COB is relatively lower, resulting in more dispersed and independent particles. These subtle differences in size, aggregation degree, and uniformity may result in different Cl^−^ removal efficiencies between COA and COB, which will be discussed next.

Based on the above analysis, COA and COB with high purity, high crystallinity, and simple preparation were selected as the reagents for Cl^−^ removal in the next experiments. The removal efficiency of Cl^−^ under different conditions is shown in Figure 2. From Figure 2a, it can be seen that with temperature increased from 20 to 80 °C, the residual Cl^−^ concentrations for COA and COB remained around 250 and 150 mg/L, respectively. According to previous studies, the removal efficiency of Cl^−^ by Cu^2+^ and Cu^0^ was improved with the increase in temperature, because of the more thorough comproportionation between Cu^2+^ and Cu^0^ under higher temperature [27]. In this work, the dissolution of Cu_2_O replaced the original comproportionation, thus significantly reducing the reaction temperature. As shown in Figure 2b, with the multiple of theoretical reagent dosage (MTRD) increased from 2 to 12, the residual Cl^−^ concentrations for COA and COB were gradually decreased from 405.5 to 201.9 mg/L and from 234.5 to 92.8 mg/L, respectively. When MTRD was further increased to 20, the reduction in residual Cl^−^ concentration for COA and COB became less obvious. To keep energy consumption and reagent costs at a lower level, next experiments were conducted under room temperature and MTRD of 12.

The effect of H_2_SO_4_ concentration is shown in Figure 2c, from which it can be observed that the concentration of Cl^−^ decreased sharply with increasing H_2_SO_4_ concentration. Under neutral conditions, the residual Cl^−^ concentration was above 400 mg/L, due to the relatively poor solubility of Cu_2_O. As the H_2_SO_4_ concentration increased from 25 to 150 g/L, the residual concentrations of Cl^−^ for COA and COB decreased from 398.0 to 195.1 mg/L and from 240.8 to 38.5 mg/L, respectively. In view of this, it can be determined that H_2_SO_4_ concentration was an extremely critical factor for Cl^−^ removal by Cu_2_O. As shown in Figure 2d, with the initial Cl^−^ concentration increased from 1 to 6 g/L, the residual Cl^−^ concentration for COA increased sharply from 0.498 to 2.948 g/L. In contrast, the residual concentration of Cl^−^ for COB was always below 100 mg/L. Based on the above data, it can be concluded that the performance of COB in Cl^−^ removal was superior to that of COA, likely due to the low aggregation and high uniformity of COB. Therefore, COB prepared via the liquid-phase reduction method was ultimately selected as the appropriate reagent in this work.

From Figure 2e, it can be seen that 25 min was sufficient for COB to reduce the residual Cl^−^ concentration to below 100 mg/L. At a reaction time of 120 min, the residual concentration of Cl^−^ reached 61.9 mg/L. The residual Cu^2+^ concentration as a function of time is shown in Figure 2f, from which it can be observed that the residual Cu^2+^ concentration increased with increasing reaction time. At 25 min and 120 min, the residual Cu^2+^ concentration, respectively, reached high values of 1.07 and 1.79 g/L. In practical applications, the treatment time can be adjusted according to the requirements of different enterprises. In addition, the high residual Cu^2+^ concentration inevitably caused secondary pollution and reagent waste. Improvements to reduce the residual Cu^2+^ concentration will be discussed next.

### 3.2. Solid Products for the Removal of Cl^−^ by COA and COB

After the reaction parameters were determined, the composition of the solid products for COA and COB was explored. From the strong and sharp peaks at 2-Theta of 28.5°, 47.4°, and 56.2° in the XRD spectra (Figure 3a), the major component of the obtained solids was determined to be CuCl. Additionally, a small quantity of Cu_2_O, corresponding to signals at 2-Theta of 36.4° and 42.3°, remained after the reactions. In view of the rather high reactivity of Cu_2_O with H^+^, it was hypothesized that the generated CuCl covered the surface of undissolved Cu_2_O, limiting the contact between undissolved Cu_2_O and Cl^−^ and their further reaction. Strong signals at 2-Theta of 43.1° and 50.2° also indicate that a certain quantity of Cu^0^ also formed during the treatment. For the effective removal of Cl^−^, the dosed Cu_2_O (or the dissolved Cu^+^) was in excess. According to previous reports, Cu^+^ in acidic aqueous conditions can be oxidized to Cu^2+^ [32,33,34] or disproportionate to produce Cu^+^ and Cu^0^ [35]. Based on the rather high residual concentration of Cu^2+^ and the existence of Cu^0^, it can be determined that both of the above two reaction pathways existed, providing a basis for reducing the residual Cu^2+^ concentration in subsequent work.

Considering that signals for the extremely trace substances may be missed by XRD analysis, XPS analysis was further conducted to determine the elemental valence of the obtained solids. From the XPS-SUM results in Figure 3b, it can be observed that the spectra for the residual solids from COA and COB are similar, indicating the same elemental composition. As shown in the XPS-Cl2p spectrum for the products of COA in Figure 3c, the signals at binding energy (E_B_) of 198.6–198.9 eV and 200.2–200.5 eV correspond, respectively, to the 2p3/2 and 2p1/2 orbits of CuCl [36,37], indicating to some extent that the chloride-containing product was solely CuCl. From the XPS-Cu2p spectra in Figure 3d, it can be determined that the signals at E_B_ of 931.9 eV and 951.9–952.0 eV correspond, respectively, to the 2p3/2 and 2p1/2 orbits of Cu^+^, according to the NIST XPS database. Additionally, the peaks at E_B_ of 932.5–932.6 eV and 952.5–952.7 eV, respectively, correspond to the 2p3/2 and 2p1/2 orbits of Cu^0^ [38,39]. The XPS-CuLM2 spectra are shown in Figure 3e, where the Auger parameters (α) for signals at E_B_ of 568.1–568.8 eV and 570.9–571.1 eV were, respectively, calculated to be 1850.6–1851.0 eV for Cu^0^ and 1847.4–1847.6 eV for Cu^+^, according to Equations (4) and (5) [40,41,42,43]. Based on the above XRD and XPS analyses, it can be determined that the residual solids from both COA and COB were composed of CuCl, Cu_2_O, and Cu^0^.(4)$$\mathrm{hv}=\;E_{K(\mathrm{LM}2)}+\;E_{B(\mathrm{LM}2)}=1486.6\;\mathrm{eV}$$(5)$$\alpha_{(\mathrm{Cu}2p3/2)}\;=\;E_{B(\mathrm{Cu}2p3/2)}\;+\;E_{K(\mathrm{LM}2)}$$
where hv is the excitation source energy, E_k_ is the kinetic energy of Auger electrons.

Morphological characterization results for the solid products of COA and COB are shown in Figure 4. The solid products of COA underwent significant agglomeration (Figure 4a), signifying poor dispersibility. Additionally, irregular shapes and rough surfaces can also be observed in Figure 4b. By contrast, the solid products of COB appear more dispersed according to Figure 4f; moreover, some cylindrical-shaped particles can be seen Figure 4g, indicating that the solid products of COB exhibited more regular shapes than those of COA. As shown in the SEM-EDS results (Figure 4c–e,h,i), for the solid products of COA and COB, the contents for the copper, chlorine, and oxygen elements were, respectively, 51.08%, 30.75%, 18.17% and 53.01%, 39.12%, 7.87%, generally consistent with the results in Figure 2 and Figure 3.

### 3.3. VC-Improved Removal of Cl^−^ by Cu_2_O

As mentioned above, further improvement in the reduction in residual Cu^2+^ concentration was necessary for the application of this new method. In view of the speculation that it was the disproportionation and oxidation reactions of Cu^+^ that led to the high residual concentration of Cu^2+^, a solution of adding an appropriate amount of reductive VC simultaneously with Cu_2_O was proposed. From Figure 5a, it can be observed that the addition of VC indeed helped reduce the dosage of Cu_2_O. Under a VC dosage of 0.10 g, the residual Cl^−^ concentration at MTRDs of 2, 3, 4, and 5, respectively, reached 97.5, 53.8, 43.2, and 40.7 mg/L. Furthermore, with increasing VC dosage, the residual concentration of Cl^−^ exhibited a trend of an initial rapid decrease followed by a gradual increase, which will be further discussed in conjunction with subsequent experimental results. In Figure 5b, it can be clearly observed that the residual concentration of Cu^2+^ was sharply decreased to a value below 50 mg/L with the VC dosage enhanced to 0.10 g. Overall, an MTRD of 3 and a VC dosage of 0.10 g appear more appropriate, with low residual concentrations of both Cl^−^ and Cu^2+^ at 53.8 and 18.4 mg/L, respectively, in this study. In industrial practice, the dosages of Cu_2_O and VC can be further adjusted based on treatment needs.

To evaluate the reproducibility of the COB synthesis route, three batches of COB were prepared using the same procedures. The Cl^−^ removal results show that the Cl^−^ removal efficiency of these three COB samples remained at approximately 90%. These findings indicate that the COB synthesis method adopted in this study exhibits good reproducibility and possesses potential for practical applications.

The XRD results for the solid products with the addition of VC are shown in Figure 6, where it can be observed that the major composition was still CuCl, Cu_2_O, and Cu^0^. With the VC dosage increased from 0 to 0.10 g, the yield of CuCl increased significantly, consistent with the observation that the appropriate addition of VC enhanced Cl^−^ removal efficiency. Moreover, the amount of Cu^0^ generated with the addition of VC was much greater than that generated without VC, indicating that a fraction of Cu^2+^ and Cu^+^ was inevitably reduced to Cu^0^. When the VC dosage was further increased from 0.10 to 0.30 g, the yields of CuCl and Cu^0^ exhibited downward and upward trends, respectively.

To further determine the transformation rule for Cu^2+^, Cu^+^, and Cu^0^ with the addition of VC, UV-vis spectra for the reaction system were obtained and are shown in Figure 7. Figure 7a shows the variation in UV-vis spectra with the addition of VC at different reaction times. At the initial reaction stage (1 min), the signal for Cu^+^ at a wavelength of 808 nm was relatively low. As the reaction proceeded to 3 and 9 min, peaks at a wavelength of 808 nm became stronger, indicating the increased concentration of Cu^+^. At 9 min, VC was added into the reaction solution, immediately after which the signal at a wavelength of 808 nm was significantly increased. Thus, it can be determined that the addition of VC indeed contributed to the generation of Cu^+^. From Figure 7b–d, it can be observed that the signals at a wavelength of 808 nm with the addition of 0.1 g VC were much stronger than those without VC, which explains the high Cl^−^ removal efficiency observed in Figure 5a. Furthermore, the signal intensities for Cu^+^ with the addition of 0.3 g VC were close to those with 0.1 g VC, indicating that a portion of VC did not contribute to the generation of Cu^+^ at excessively high VC dosages.

In view of the above results, the impact of the addition of VC on the removal of Cl^−^ can be explained according to Figure 8. In cases where only overstoichiometric Cu_2_O was dosed, Cu_2_O dissolved immediately under acidic conditions, releasing Cu^+^ (Equation (6)). Subsequently, a portion of Cu^+^ combined with Cl^−^ to generate CuCl (Equation (2)), leading to Cl^−^ removal, simultaneously, surplus Cu^+^ was oxidized by oxygen (O_2_) to Cu^2+^ or disproportionated to produce Cu^2+^ and Cu^0^ (Equations (7) and (8)). With a low dosage of VC added simultaneously with Cu_2_O, the strongly reductive VC helped inhibit the oxidation of Cu^+^ by O_2_ and additionally reduced Cu^2+^ generated from disproportionation back to Cu^+^ (Equation (9)), thereby improving Cl^−^ removal efficiency while reducing Cu_2_O dosage. In contrast, at excessively high VC dosages, excessive VC can reduce a portion of Cu^2+^ and Cu^+^ to form Cu^0^ according to Equations (10) and (11).

During the treatment, VC was oxidized to generate dehydroascorbic acid (DHA), which may be transferred to diketogulonic acid (KGA) over an extended period. It has been reported that VC, DHA, and KGA exhibit little toxicity, signifying that little adverse effect can be induced by these residual substances on the utilization or discharge of the treated wastewater. Additionally, thermodynamic analysis and the literature reports indicate that dissolved oxygen (DO) in wastewater can undergo redox reactions with VC, resulting in partial consumption of VC before it reacts with the target pollutant. To mitigate the negative impact of DO, the following measures can be adopted in practical applications. (1) Performing N_2_ stripping prior to VC addition to reduce the DO concentration to below 2 mg/L. (2) Adjusting the VC dosage according to the DO concentration. (3) Applying a staged dosing strategy to reduce the contact time between VC and DO. These strategies need to be validated in pilot/industrial applications before being adopted.

The removal of Cl^−^ relies on the core reaction in which Cu^+^ combines with Cl^−^ to form CuCl. VC serves to maintain the concentration of Cu^+^ by reducing Cu^2+^ and inhibiting the oxidation of Cu^+^. However, excessive VC disrupts this balance. Furthermore, excessive VC results in the ineffective consumption of two key reagents, thereby increasing the process cost. In addition, the ideal byproduct of this process is high-purity CuCl, whereas Cu^0^ generated by excessive VC becomes an impurity waste, leading to an increase in the total amount of solid waste. Therefore, in practical applications, the dosage of VC must be strictly controlled to avoid the multiple adverse effects caused by excess.(6)$${\mathrm{Cu}}_{2}O+{\;2H}^{+}\;\to {\;2\mathrm{Cu}}^{+}+{\;H}_{2}O$$(7)$${4\mathrm{Cu}}^{+}+{\;O}_{2}+{4H}^{+}\;\to \;4{\mathrm{Cu}}^{2+}+{2H}_{2}O$$(8)$${2\mathrm{Cu}}^{+}\;\to \;{\mathrm{Cu}}^{2+}+{\mathrm{Cu}}^{0}$$(9)$${2\mathrm{Cu}}^{2+}+{\;C}_{6}H_{8}O_{6\;}\to \;{2\mathrm{Cu}}^{+}+C_{6}H_{6}O_{6\;}+{\;2H}^{+}$$(10)$${2\mathrm{Cu}}^{+}+{\;C}_{6}H_{8}O_{6\;}\to {\;2\mathrm{Cu}}^{0}+{\;C}_{6}H_{6}O_{6\;}+{2H}^{+}$$(11)$${\mathrm{Cu}}^{2+}+{\;C}_{6}H_{8}O_{6}\;\to {\;\mathrm{Cu}}^{0}+C_{6}H_{6}O_{6}+{2H}^{+}$$

According to the preparation procedures of COB, it can be calculated that approximately 3333 g of NaOH, 3472 g of CuSO_4_·5H_2_O, and 1222 g of VC are needed to prepare 1 kg of COB, with the reagent cost of approximately 89 rmb. Additionally, for the treatment of 1 ton of wastewater containing 500 mg/L of Cl^−^ and 100 g/L of H_2_SO_4_, the reagent cost of this proposed method will be nearly 290 rmb. Furthermore, the generated CuCl can be regarded as a kind of byproduct, which can further lower the treatment cost. In contrast, for the conventional copper slag method, treating 1 ton of wastewater of the same composition would require 1803 g of copper powder and 21,127 g of CuSO_4_·5H_2_O, with a rather high cost of approximately 421 rmb [27]. Additionally, the residual Cl^−^ concentration in this study was below 50 mg/L and met the relevant technical specification formulated by the China National Resources Recycling Association. However, the Cl^−^ removal efficiency of the traditional copper slag is relatively limited with the residual Cl^−^ concentration always above 150 mg/L [27]. The extremely high residual Cu^2+^ concentration exceeding 20,000 mg/L also poses challenges for the subsequent treatment of wastewater. Therefore, the proposed method demonstrates clear advantages compared with conventional approaches in terms of Cl^−^ removal efficiency, cost, environmental safety, and ease of operation.

### 3.4. Treatment of the Actual Wastewater

Ultimately, to further evaluate the performance of the proposed method for Cl^−^ removal, a series of experiments were conducted in the actual wastewater, and the results are shown in Table 2. Given the complexity of Cl^−^-containing wastewater systems, three types of actual wastewater were employed in this study, namely the electrolytic wastewater from the chlor-alkali industry, the desulfurization wastewater from the copper smelter, and the metal surface polishing wastewater from the steel works. Due to the high difficulty in homogenizing large-volume wastewater, the reaction time was extended to 2 h and the MTRD was increased to 3.3. Notably, the metal surface polishing wastewater contained a high concentration of Fe^3+^, which could potentially consume VC. Consequently, the VC dosage was set higher than the previously determined value. After treatment, the concentration of Cl^−^ in the electrolytic wastewater was reduced from 2752 to 189.6 mg/L, where the slightly elevated Cl^−^ residual concentration might be attributed to the low wastewater acidity. For the desulfurization wastewater, Cl^−^ was effectively removed, with its concentration decreased from 3846 to 78.3 mg/L, thanks to the high acidity. Additionally, the removal of Cl^−^ from the metal surface polishing wastewater seems less desirable with a residual concentration of 393.7 mg/L. Although this wastewater exhibited high acidity, the extremely high initial Cl^−^ concentration posed certain challenges for treatment. These results also indicate that the influence of co-existing pollutants, including SO_4_^2−^, NO_3_^−^, F^−^, Al^3+^, Fe^3+^, etc., on the removal of Cl^−^, can be ignored.

The residual Cu^2+^ concentration in these three types of wastewaters reached 27.5, 18.3, and 22.6 mg/L, respectively. To avoid secondary pollution and reagent waste, H_2_S (0.01 L/min, 10 min) and N_2_ (0.01 L/min, 30 min) gases were sequentially introduced into the treated wastewater, leading to an ultimate Cu^2+^ concentration below 1 mg/L, which met certain typical discharge standards in China and internationally. The produced copper sulfide (Cu_2_S_3_) can be utilized as a kind of byproduct in the fields of smelting, catalysis, semiconductor manufacturing, etc. After treatment in this study, the fate of copper includes unreacted Cu_2_O, CuCl product, and Cu_2_S_3_ byproduct. Indeed, CuCl can be transformed into Cu_2_O through a straightforward solid- or liquid-phase oxidation process, thereby allowing the regeneration and reuse of the Cl^−^ removal agent. This aspect will be addressed in subsequent studies.

## 4. Conclusions

This work systematically investigated the performance of the novel VC-assisted Cu_2_O method for Cl^−^ removal from strongly acidic wastewater and validated the feasibility of this method. Liquid-phase reduction was selected as the synthesis method for Cu_2_O based on product purity and composition. Systematic parameter optimization revealed that the increased reagent dosage and acidity significantly enhanced the Cl^−^ removal efficiency, while the influence of other factors was less significant. With the assistance of 0.10 g VC, the removal of Cl^−^ by Cu_2_O was significantly improved: the MTRD decreased from 12 to 3, the residual Cu^2+^ concentration decreased from 1197 to 18.4 mg/L, and the residual Cl^−^ concentration decreased from 88.4 to 53.8 mg/L, with other parameters kept constant. After treatment without VC, the dosed Cu_2_O existed in four forms, including CuCl, Cu^2+^, Cu^0^, and Cu_2_O, among which the generation of Cu^2+^ and Cu^0^, through oxidation and disproportionation of Cu^+^, served as the main reason for the unsatisfactory Cl^−^ removal efficiency. Fortunately, VC is precisely capable of inhibiting both the oxidation and disproportionation of Cu^+^, thus leading to the improved removal of Cl^−^ by Cu_2_O. Before the large-scale application of this proposed method. The relatively poor stability of Cu_2_O remains a big issue to be solved. Preparing fresh Cu_2_O prior to wastewater treatment may be a feasible approach. After treatment in this study, the fate of copper includes unreacted Cu_2_O, CuCl product, and Cu_2_S_3_ byproduct. The regeneration and reuse of Cu_2_O from CuCl will be addressed in the subsequent studies.

## Figures and Tables

**Figure 1 toxics-13-00890-f001:**
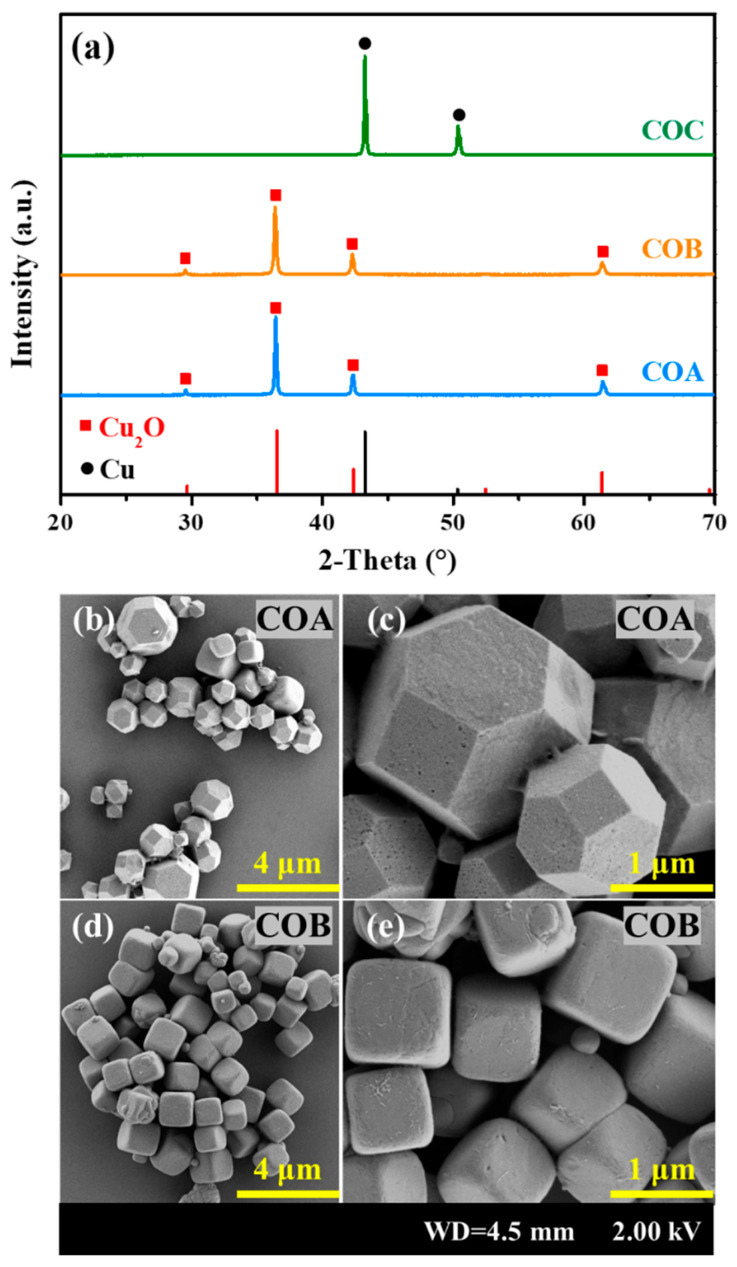
XRD spectra for COA, COB, and COC samples (**a**), and SEM results for COA (**b**,**c**) and COB (**d**,**e**).

**Figure 2 toxics-13-00890-f002:**
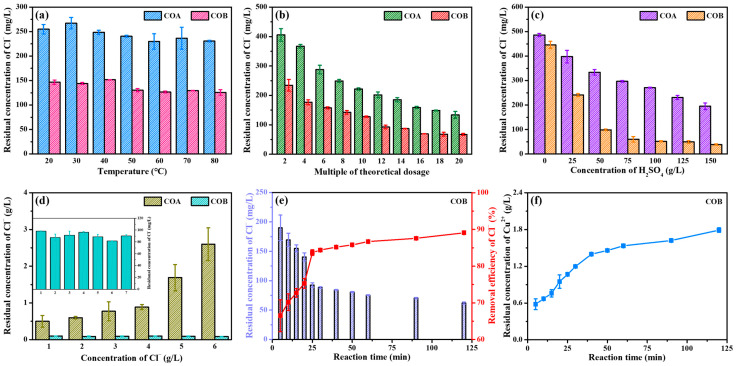
Residual concentration of Cl^−^ as a function of temperature (**a**), multiple of theoretical dosage (**b**), concentration of H_2_SO_4_ (**c**), concentration of Cl^−^ (**d**), and reaction time (**e**), and residual concentration of Cu^2+^ as a function of reaction time (**f**) after treatment by COA and COB. Error bars represent the standard deviation (SD) from triplicate measurements (*n* = 3).

**Figure 3 toxics-13-00890-f003:**
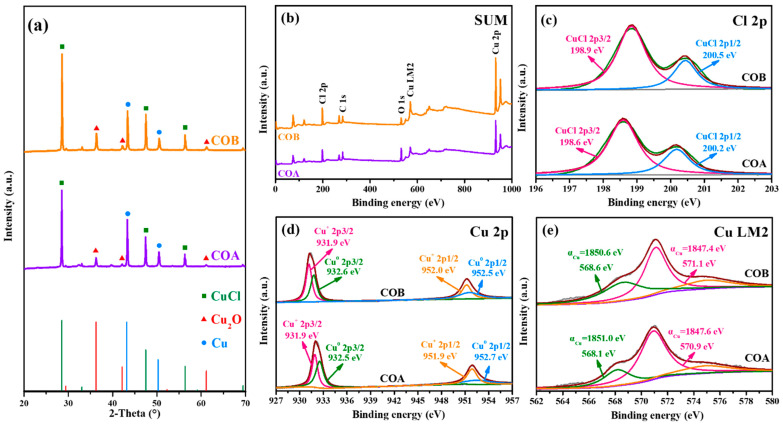
XRD (**a**), XPS-SUM (**b**), XPS-Cl2p (**c**), XPS-Cu2p (**d**), and XPS-CuLM2 (**e**) spectra for the solid products of COA and COB.

**Figure 4 toxics-13-00890-f004:**
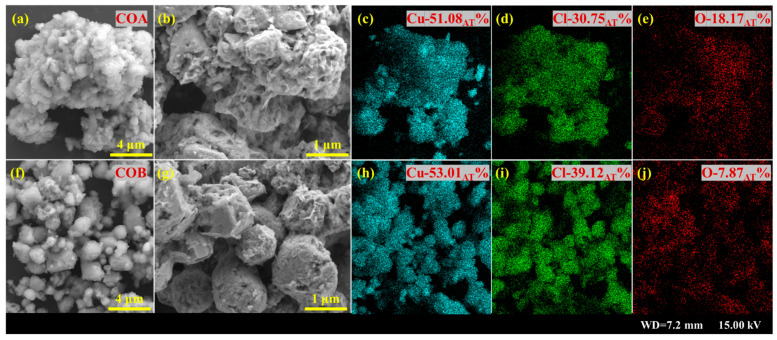
SEM results for the solid products of COA and COB (**a**,**b**,**f**,**g**), and SEM-EDS results for copper (**c**,**h**), chloride (**d**,**i**), and oxygen (**e**,**j**) elements in the solid products.

**Figure 5 toxics-13-00890-f005:**
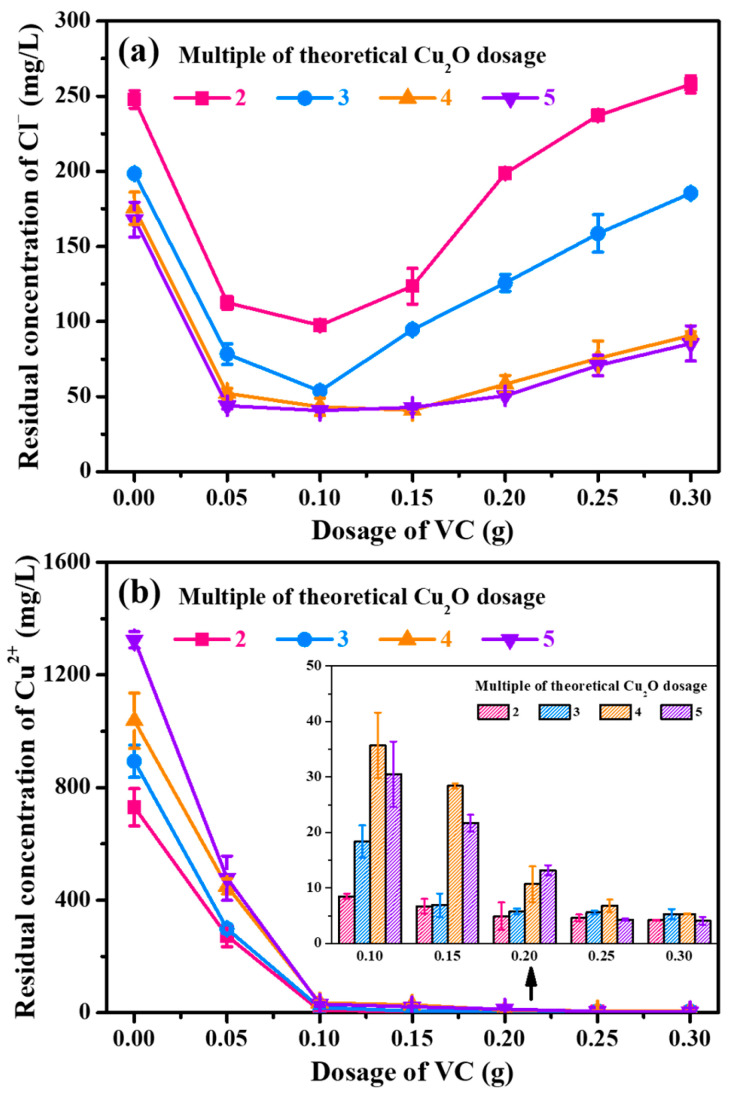
Residual concentrations of Cl^−^ (**a**) and Cu^2+^ (**b**) as a function of VC dosage after treatment by VC-assisted COB. Error bars represent the SD from triplicate measurements (*n* = 3).

**Figure 6 toxics-13-00890-f006:**
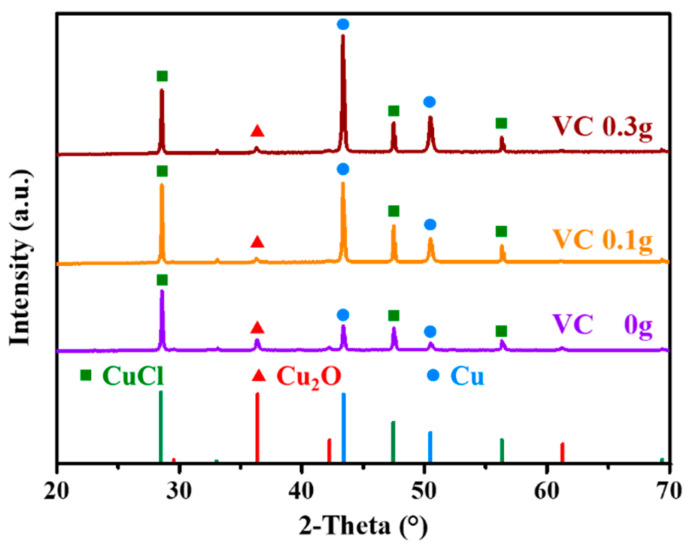
XRD spectra for the solid products of VC-assisted COB.

**Figure 7 toxics-13-00890-f007:**
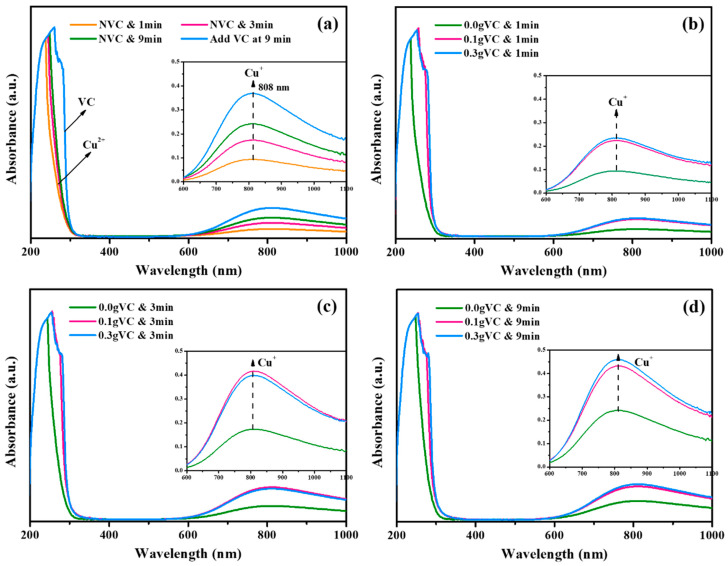
UV-vis absorption spectra for the reaction solutions with VC added halfway (**a**), and at the beginning after the reaction time of 1 min (**b**), 3 min (**c**), and 9 min (**d**).

**Figure 8 toxics-13-00890-f008:**
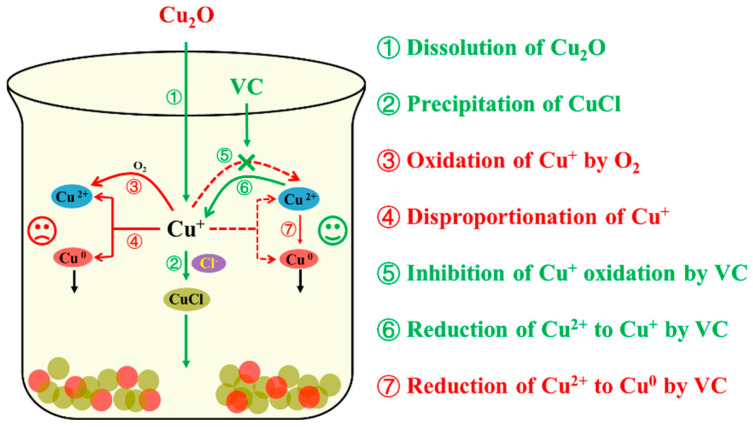
Sketch map for the removal mechanism of Cl^−^ by Cu_2_O and VC.

**Table 1 toxics-13-00890-t001:** Composition of the actual wastewater.

Indicator	Concentration		
	A	B	C
Acid type	HCl	H_2_SO_4_	HNO_3_ & HCl
pH value	1.31	0.03	<0
[Cl^−^]	2752 mg/L	3846 mg/L	13,570 mg/L
[SO_4_^2−^]	1360 mg/L	49,200 mg/L	/
[F^−^]	/	543 mg/L	/
[NO_3_^−^]	/	/	21,240 mg/L
[Cu^2+^]	/	67 mg/L	/
[Fe^3+^]	32 mg/L	/	3846
[Al^3+^]	/	/	1560
[Heavy metals]	/	680 mg/L	/
COD	320 mg/L	/	/

Note: A, B, and C, respectively, represent the electrolytic wastewater from the chlor-alkali industry, the desulfurization wastewater from the copper smelter and the metal surface polishing wastewater from the steel works.

**Table 2 toxics-13-00890-t002:** Actual wastewater treatment results.

Indicator	A	B	C
Calculated Cu_2_O dosage (g)	84	117	413
Actual Cu_2_O dosage (g)	92	129	454
VC dosage (g)	28	38	163
Reaction time (h)	2		
Residual concentration of Cl^−^ (mg/L)	189.6	78.3	393.7
Removal efficiency of Cl^−^	93.1%	98.0%	97.1%
Residual concentration of Cu^2+^ (mg/L)	27.5	18.3	22.6

## Data Availability

The datasets used and/or analyzed during the current study are available from the corresponding author on reasonable request.

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
