# Peer review of "Removal Efficiency and Mechanism for Cl− from Strongly Acidic Wastewater by VC-Assisted Cu2O: Comparison Between Synthesis Methods"

_toxics, 2025, doi:10.3390/toxics13100890_

Round 1
Reviewer 1 Report
Comments and Suggestions for Authors
The manuscript entitled “Removal Efficiency and Mechanism for Cl⁻ from Strongly Acidic Wastewater by VC-Assisted Cu₂O: Comparison Between Synthesis Methods” presents a novel approach for chloride removal in strongly acidic wastewater using cuprous oxide (Cu₂O) and ascorbic acid (VC) as an enhancer. The study addresses a significant environmental problem, as high concentrations of chloride ions in acidic effluents are challenging to remove and cause equipment corrosion and environmental degradation. The authors provide a thorough experimental design, comparing different Cu₂O synthesis methods, testing multiple parameters (dosage, acidity, reaction time, etc.), and applying the method to actual industrial wastewater. The integration of mechanistic studies (XRD, SEM-EDS, XPS, UV-Vis) adds strength to the manuscript. However, some aspects require clarification, deeper discussion, or additional data to support the conclusions. Below, I highlight the main strengths, weaknesses, and open questions. The manuscript presents promising and relevant findings, but it requires major revisions before acceptance. Specifically, the authors should: Provide discussion on Cu²⁺ management and environmental trade-offs. Include at least a preliminary cost or feasibility analysis. Expand discussion on scalability and industrial application. Clarify statistical robustness of their results.
- While Cl⁻ removal is demonstrated, the generation of Cu²⁺ as a byproduct raises concerns about secondary pollution. The manuscript should discuss strategies for Cu²⁺ recovery or safe disposal.
- Stability of Cu₂O/VC under continuous or large-scale operations is not evaluated. Scalability issues should be addressed.
- Although reagent reduction is mentioned, no cost analysis (e.g., Cu₂O vs. conventional methods) is provided.
- The introduction reviews alternatives, but the discussion does not sufficiently compare the proposed method’s efficiency, cost, and environmental safety against established approaches.
- The removal process is described experimentally, but kinetic data or reaction models are not provided, which could help generalize the applicability.
- The solid byproducts (CuCl, Cu⁰, Cu₂O) could have industrial reuse potential, but this is not explored.
- While the optimum (0.1 g) is discussed, higher dosages leading to Cu⁰ formation are not fully analyzed in terms of implications for process efficiency and waste generation.
- Although some figures show error bars, the text rarely discusses reproducibility or statistical significance.
- The use of ascorbic acid is generally benign, but its fate in large-scale wastewater systems is not discussed.
- How does the cost of Cu₂O synthesis and VC addition compare with conventional Cl⁻ removal methods (e.g., lime treatment, ion exchange, membrane processes) in terms of operational expenditure?
- What strategies do you propose to mitigate the secondary Cu²⁺ pollution observed in your experiments? Could Cu²⁺ be recovered or recycled?
- Have you evaluated the stability and reusability of Cu₂O across multiple treatment cycles?
- Could the solid byproducts (CuCl, Cu⁰, Cu₂O) be valorized in industrial applications to reduce waste generation?
- What is the potential effect of co-existing ions (e.g., sulfates, nitrates, fluorides, heavy metals) in real wastewater on the Cl⁻ removal mechanism?
- How does the process scale beyond the 5 L tests reported? What challenges do you foresee in pilot or industrial-scale applications?
- Did you evaluate the kinetics of Cl⁻ removal to determine whether the reaction follows a specific order or rate-controlling step?
- How does the presence of dissolved oxygen in wastewater affect the efficiency of VC-assisted Cu₂O, given the competing oxidation pathways of Cu⁺?
- What are the environmental and safety implications of handling large amounts of Cu₂O and VC in industrial settings?
- Could alternative green reductants (e.g., plant extracts, sugars) substitute VC while maintaining efficiency?
- How reproducible are the results across multiple wastewater samples? Did you test seasonal or batch variability in industrial effluents?
- What are the regulatory perspectives does the residual Cu concentration after treatment meet typical discharge standards in China or internationally?
Author Response
Comment 1:
While Cl⁻ removal is demonstrated, the generation of Cu²⁺ as a byproduct raises concerns about secondary pollution. The manuscript should discuss strategies for Cu²⁺ recovery or safe disposal.
Response 1:
Thank you for your suggestion. The related contents haven been supplemented in lines 420-426 of the manuscript and described as follows. After the treatment of actual wastewater, the residual Cu2+ concentration in these three types of wastewaters reached 27.5, 18.3, and 22.6 mg/L, respectively. To avoid the secondary pollution and reagent waste, H2S (0.01 L/min, 10 min) and N2 (0.01 L/min, 30 min) gases were sequentially introduced into the treated wastewater, leading to an ultimate Cu2+ concentration below 1 mg/L. The produced copper sulfide (Cu2S3) can be utilized as a kind of byproduct in the fields of smelting, catalysis, semiconductor manufacturing, etc.
Comment 2:
Stability of Cu₂O/VC under continuous or large-scale operations is not evaluated. Scalability issues should be addressed.
Response 2:
Thank you for your suggestion. The related contents haven been supplemented in lines 445-448 of the manuscript. As the reviewer mentioned, the relatively poor stability of Cu2O is indeed a big issue to be solved before the large-scale application of this method. Preparing fresh Cu2O prior to wastewater treatment may be a feasible approach. In contrast, the industrial production of VC is technically mature and features stable capacity, with wide applications in food, pharmaceuticals, feed, and other fields. The key to storing VC lies in conditions such as sealing, protection from light, and control of temperature and humidity. Under standardized operations, the storage of VC is relatively easy and can meet the requirements of industrial-scale storage.
Comment 3:
Although reagent reduction is mentioned, no cost analysis (e.g., Cu₂O vs. conventional methods) is provided.
Response 3:
Thank you for your suggestion. The related contents haven been supplemented in lines 383-391 of the manuscript. According to the preparation procedures of COB, it can be calculated that approximately 3333 g of NaOH, 3472 g of CuSO4·5H2O, and 1222 g of VC are needed to prepare 1 kg of COB, with the reagent cost of approximately 89 rmb. Additionally, for the treatment of 1 ton of wastewater containing 500 mg/L of Cl− and 100 g/L of H2SO4, the reagent cost of this proposed method will be nearly 290 rmb. Furthermore, the generated CuCl and Cu2S3 can be regarded as byproducts, which can further lower the treatment cost.
Comment 4:
The introduction reviews alternatives, but the discussion does not sufficiently compare the proposed method’s efficiency, cost, and environmental safety against established approaches.
Response 4:
Thank you for your suggestion. The related contents haven been supplemented in lines 383-398 of the manuscript. According to the preparation procedures of COB, it can be calculated that approximately 3333 g of NaOH, 3472 g of CuSO4·5H2O, and 1222 g of VC are needed to prepare 1 kg of COB, with the reagent cost of approximately 89 rmb. Additionally, for the treatment of 1 ton of wastewater containing 500 mg/L of Cl− and 100 g/L of H2SO4, the reagent cost of this proposed method will be nearly 290 rmb. Furthermore, the generated CuCl can be regarded as a kind of byproduct, which can further lower the treatment cost. In contrast, for the conventional copper slag method, treating 1 ton of wastewater of the same composition would require 1803 g of copper powder and 21127 g of CuSO4·5H2O, with a rather high cost of approximately 421 rmb. Additionally, the residual Cl− concentration in this study was below 50 mg/L and met the relevant technical specification formulated by China National Resources Recycling Association. However, the Cl− removal efficiency of the traditional copper slag is relatively limited with the residual Cl− concentration always above 150 mg/L. The extremely high residual Cu2+ concentration exceeding 20000 mg/L also poses challenges for the subsequent treatment of wastewater. Therefore, the proposed method demonstrates clear advantages compared with conventional approaches in terms of Cl− removal efficiency, cost, environmental safety, and ease of operation.
Comment 5:
The removal process is described experimentally, but kinetic data or reaction models are not provided, which could help generalize the applicability.
Response 5:
Thank you for your suggestion. We fully agree with the reviewer that kinetic data and reaction modeling would strengthen the general applicability of our method. However, due to the complexity of the reaction system and the limited time available for revision, we have not been able to complete these studies. We plan to address this in our future work and will provide a detailed kinetic analysis in a follow-up publication.
Comment 6:
The solid byproducts (CuCl, Cu⁰, Cu₂O) could have industrial reuse potential, but this is not explored.
Response 6:
Thank you for your suggestion. The related contents haven been supplemented in lines 426-430 of the manuscript. After treatment in this study, the fate of copper includes unreacted Cu2O, CuCl product, and Cu2S3 byproduct. Indeed, CuCl can be transformed into Cu2O through a straightforward solid- or liquid- phase oxidation process, thereby allowing the regeneration and reuse of the Cl− removal agent. However, our current work primarily focuses on the Cl− removal efficiency and reaction mechanism, and a comprehensive investigation of byproduct utilization was not included in the present study due to time constraints. We plan to systematically evaluate the feasibility of reusing these byproducts in future research, including their characterization, purification, and potential applications in relevant industrial processes.
Comment 7:
While the optimum (0.1 g) is discussed, higher dosages leading to Cu⁰ formation are not fully analyzed in terms of implications for process efficiency and waste generation.
Response 7:
Thank you for your suggestion. The related contents haven been supplemented in lines 367-374 of the manuscript. The removal of Cl− relies on the core reaction in which Cu⁺ combines with Cl− to form CuCl. VC serves to maintain the concentration of Cu⁺ by reducing Cu2⁺ and inhibiting the oxidation of Cu⁺. However, excessive VC disrupts this balance. Furthermore, excessive VC results in the ineffective consumption of two key reagents, thereby increasing the process cost. In addition, the ideal byproduct of this process is high-purity CuCl, whereas Cu⁰ generated by excessive VC becomes an impurity waste, leading to an increase in the total amount of solid waste. Therefore, in practical applications, the dosage of VC must be strictly controlled to avoid the multiple adverse effects caused by excess.
Comment 8:
Although some figures show error bars, the text rarely discusses reproducibility or statistical significance.
Response 8:
Thank you for your suggestion. The related contents haven been supplemented in lines 218-219, 311 and 312-316 of the manuscript. The explanations of the error bars in the experimental results have been revised and improved for clarity. Additionally, to evaluate the reproducibility of COB synthesis route, three batches of COB were prepared using the same procedures. The Cl⁻ removal results show that the Cl⁻ removal efficiency of these three COB samples remained at approximately 90%. These findings indicate that the COB synthesis method adopted in this study exhibits good reproducibility and possesses potential for practical applications.
Comment 9:
The use of ascorbic acid is generally benign, but its fate in large-scale wastewater systems is not discussed.
Response 9:
Thank you for your suggestion. The related contents haven been supplemented in lines 355-359 of the manuscript. During the treatment, VC was oxidized to generate dehydroascorbic acid (DHA), which may be transferred to diketogulonic acid (DKA) over an extended period. It has been reported that all of VC, DHA and KGA exhibit little toxicity, signifying that little adverse effect can be induced by these residual substances on the utilization or discharge of the treated wastewater.
Comment 10:
How does the cost of Cu₂O synthesis and VC addition compare with conventional Cl⁻ removal methods (e.g., lime treatment, ion exchange, membrane processes) in terms of operational expenditure?
Response 10:
Thank you for your suggestion. The related contents haven been supplemented in lines 383-398 of the manuscript. For the removal of Cl⁻ from H2SO4 wastewater, copper slag precipitation method is the current conventional approach as mentioned in the manuscript. Given this, we mainly compared the operational expenditure of Cl⁻ removal using copper slag and VC assisted Cu2O. For these two methods, the reactor and operating mode used are similar, with the main difference being the type of chemicals added. Therefore, the cost of the reagents directly determines the economic feasibility of the method. According to the preparation procedures of COB, it can be calculated that approximately 3333 g of NaOH, 3472 g of CuSO4·5H2O, and 1222 g of VC are needed to prepare 1 kg of COB, with the reagent cost of approximately 89 rmb. Additionally, for the treatment of 1 ton of wastewater containing 500 mg/L of Cl− and 100 g/L of H2SO4, the reagent cost of this proposed method will be nearly 290 rmb. Furthermore, the generated CuCl can be regarded as a kind of byproduct, which can further lower the treatment cost. In contrast, for the conventional copper slag method, treating 1 ton of wastewater of the same composition would require 1803 g of copper powder and 21127 g of CuSO4·5H2O, with a rather high cost of approximately 421 rmb. Therefore, the proposed method demonstrates a clear advantage compared with conventional approaches in terms of operational expenditure.
Comment 11:
What strategies do you propose to mitigate the secondary Cu²⁺ pollution observed in your experiments? Could Cu²⁺ be recovered or recycled?
Response 11:
Thank you for your suggestion. The related contents haven been supplemented in lines 420-430 of the manuscript. The residual Cu2+ concentration in these three types of wastewaters reached 27.5, 18.3, and 22.6 mg/L, respectively. To avoid the secondary pollution and reagent waste, H2S (0.01 L/min, 10 min) and N2 (0.01 L/min, 30 min) gases were sequentially introduced into the treated wastewater, leading to an ultimate Cu2+ concentration below 1 mg/L. The produced copper sulfide (Cu2S3) can be utilized as a kind of byproduct in the fields of smelting, catalysis, semiconductor manufacturing, etc.
Comment 12:
Have you evaluated the stability and reusability of Cu₂O across multiple treatment cycles?
Response 12:
Thank you for your suggestion. The related contents haven been supplemented in lines 426-430 of the manuscript. According to the SEM images of Cl⁻ removal products in the manuscript, massive CuCl accumulated on the surface of unreacted Cu2O, thus preventing further contact between Cu2O and the reaction system. We also conducted a Cl⁻ removal experiment using the product without any treatment and found that its Cl⁻ removal efficiency was very poor. This indicates that the product must undergo regeneration treatment before it can be reused in subsequent cycles. However, our current work primarily focuses on the Cl− removal efficiency and reaction mechanism, and a comprehensive investigation of reagent regeneration was not included in the present study due to time constraints. We plan to systematically evaluate the feasibility of reusing the Cl− removal products in future research, including their characterization, regeneration, and reuse value.
Comment 13:
Could the solid byproducts (CuCl, Cu⁰, Cu₂O) be valorized in industrial applications to reduce waste generation?
Response 13:
Thank you for your suggestion. The related contents haven been supplemented in lines 420-430 and of the manuscript. After treatment in this study, the fate of copper includes unreacted Cu2O, CuCl product, Cu2S3 byproduct and potential undesired Cu0. Cu2O and CuCl: After the removal of Cl−, Cu2O and CuCl will be separated together through filtration. Indeed, CuCl can be transformed into Cu2O through a straightforward solid- or liquid- phase oxidation process, thereby allowing the regeneration and reuse of the Cl− removal agent. Cu2S3 byproduct: The residual Cu2+ concentration in these three types of wastewaters reached 27.5, 18.3, and 22.6 mg/L, respectively. To avoid the secondary pollution and reagent waste, H2S (0.01 L/min, 10 min) and N2 (0.01 L/min, 30 min) gases were sequentially introduced into the treated wastewater, leading to an ultimate Cu2+ concentration below 1 mg/L. The produced Cu2S3 can be utilized as a kind of byproduct in the fields of smelting, catalysis, semiconductor manufacturing, etc. Potential undesired Cu0: Due to the addition of VC during Cl− removal, a small amount of Cu0 will be generated. On the one hand, the amount of Cu0 generated during Cl− removal can be minimized by strictly controlling the dosage of VC. On the other hand, under the conditions for straightforward solid- or liquid- phase oxidation process, a conventional method for the transformation of CuCl into Cu2O, Cu0 exhibits high stability and will not take part in the conversion. Given this, a small amount of Cu0 will exist in the regenerated Cu2O. With the increase in the number of reuse cycles of Cu2O, it is necessary to periodically add fresh Cu2O to maintain high efficiency.
However, our current work primarily focuses on the Cl− removal efficiency and reaction mechanism, and a comprehensive investigation of byproduct utilization was not included in the present study due to time constraints. We plan to systematically evaluate the feasibility of reusing these byproducts in future research, including their characterization, purification, and potential applications in relevant industrial processes.
Comment 14:
What is the potential effect of co-existing ions (e.g., sulfates, nitrates, fluorides, heavy metals) in real wastewater on the Cl⁻ removal mechanism?
Response 14:
Thank you for your suggestion. The related contents haven been supplemented in lines 407-409 of the manuscript. In this approach, Cl⁻ is removed via the specific precipitation reaction between Cl⁻ and Cu+. Given the chemical characteristics of the common co-existing ions, including sulfates, nitrates, fluorides, and heavy metals, the precipitation of Cl⁻ with Cu+ will be hardly affected. However, the ions with oxidizing properties may consume VC, resulting in a decrease in Cl⁻ removal efficiency. Therefore, when performing Cl⁻ removal of actual wastewater, the dosage of VC should be adjusted according to the wastewater composition. The actual wastewater C used in this study contained a high concentration of Fe3+, which could potentially consume VC. Consequently, the VC dosage was set higher than the previously determined value, resulting in a Cl⁻ removal efficiency of 97.1%. This indicates that it is feasible to eliminate the impact of co-existing oxidizing ions by controlling the VC dosage.
Comment 15:
How does the process scale beyond the 5 L tests reported? What challenges do you foresee in pilot or industrial-scale applications?
Response 15:
Thank you for your suggestion. Regarding the scalability of the process beyond the 5 L tests and potential challenges in pilot/industrial-scale applications, we have systematically analyzed the key considerations and proposed targeted strategies, as detailed below.
- Process scaling strategy beyond 5 L tests
The scaling-up of Cl⁻ removal process follows the geometric similarity principle and parameter consistency principle to ensure the reproducibility of lab-scale performance at larger scales.
Reactor Design: For pilot-scale tests, a continuous stirred-tank reactor with a modified impeller structure can be taken into consideration. This design maintains the same stirring intensity and liquid retention time as the 5 L system, avoiding mass transfer limitations caused by uneven mixing in larger volumes.
Key Parameter Control: The core operational parameters will be kept consistent. For industrial-scale systems, we will add a pre-mixing unit for Cu2O and wastewater to ensure rapid dispersion of the reagent before entering the main reaction tank.
Staged Scaling Plan: We plan to implement a three-stage scaling process, for instance, 5 L (lab) → 200 L (pilot) → 10 m3 (semi-industrial). Each stage will verify Cl⁻ removal efficiency, reagent utilization rate, and byproduct separation efficiency to optimize parameters for the next scale-up.
- Foreseen challenges and mitigation measures in pilot/industrial-scale applications
The foreseen challenges and mitigation measures in pilot/industrial-scale applications are listed in Table R1.
Table R1. Foreseen challenges and mitigation measures in pilot/industrial-scale applications.
|
Challenge category |
Specific challenge |
Mitigation measure |
|
Mass transfer limitation |
Uneven mixing in large reactors reduces contact between Cu2O and Cl⁻, lowering efficiency. |
1. Using multi-impeller CSTR or loop reactor to enhance radial/axial mixing. 2. Adding a static mixer in the feed pipeline to pre-disperse Cu2O. |
|
Reagent utilization and cost |
Industrial-scale operation requires large Cu2O/VC dosage, increasing operational expenditure. |
1. Integrating a CuCl regeneration unit. 2. Optimizing VC dosage via online ORP monitoring to avoid excess addition. |
|
Byproduct separation and disposal |
Large volumes of CuCl sludge are generated, increasing solid waste treatment costs. |
1. Adopting a two-stage separation process of primary centrifugation and secondary membrane filtration. 2. Collaborating with copper smelters to reuse CuCl as a raw material. |
|
Fluctuations in wastewater composition |
Industrial wastewater has variable Cl⁻/oxidizing ion concentrations, destabilizing performance. |
1. Installing an online analytical system to real-time monitor wastewater composition. 2. Equipping an automatic reagent dosing system to adjust Cu2O/VC dosage based on real-time data. |
|
Energy consumption |
Stirring and centrifugation in large-scale systems consume high energy. |
1. Using frequency-adjustable motors for stirrers to match mixing intensity with wastewater viscosity. 2. Adopting low-speed decanter centrifuges to reduce energy consumption. |
In summary, the proposed scaling strategy and mitigation measures address the core issues of reproducibility, cost-effectiveness, and stability in pilot/industrial applications. We plan to validate the pilot-scale system in our following research to provide further data support for industrialization.
Comment 16:
Did you evaluate the kinetics of Cl⁻ removal to determine whether the reaction follows a specific order or rate-controlling step?
Response 16:
Thank you for your suggestion. We fully agree with the reviewer that kinetics of Cl⁻ removal would strengthen the general applicability of our method. However, due to the complexity of the reaction system and the limited time available for revision, we have not been able to complete these studies. We plan to address this in our future work and will provide a detailed kinetic analysis in a follow-up publication.
Comment 17:
How does the presence of dissolved oxygen in wastewater affect the efficiency of VC-assisted Cu₂O, given the competing oxidation pathways of Cu⁺?
Response 17:
Thank you for your suggestion. The related contents haven been supplemented in lines 359-366 of the manuscript. Although this study did not specifically include a DO gradient experiment, thermodynamic analysis and literature reports indicate that DO in wastewater can undergo redox reactions with VC, resulting in partial consumption of VC before it reacts with the target pollutant. Under typical industrial wastewater conditions with DO concentrations ranging from 4–8 mg/L, the oxidation rate of VC can be significantly enhanced, particularly in the presence of transition metal ions such as Fe3+ and Cu2+. To mitigate the negative impact of DO, the following measures can be adopted in practical applications. (1) Performing N2 stripping prior to VC addition to reduce the DO concentration to below 2 mg/L. (2) Adjusting the VC dosage according to the DO concentration. (3) Applying a staged dosing strategy to reduce the contact time between VC and DO. These strategies need to be validated in pilot/industrial applications before being adopted.
Comment 18:
What are the environmental and safety implications of handling large amounts of Cu₂O and VC in industrial settings?
Response 18:
Thank you for your suggestion. Environmental and safety implications of handling Cu2O and VC at industrial scale are as follows.
Environmental considerations of Cu2O and VC: Cu2O is toxic to aquatic life if released. Closed-loop handling and recovering CuCl byproduct for reuse can be taken into consideration to minimize discharge. VC is biodegradable but may cause eutrophication at high concentrations. Pretreating wastewater to remove excess VC before discharge can reduce this risk.
Safety considerations of Cu2O and VC: Cu2O dust can cause respiratory, skin, and eye irritation. Using dust control, wearing personal protective equipment (masks, gloves, and goggles), and storing Cu2O in sealed containers can improve safety. Large quantities of stacked VC are combustible and weakly acidic. They should be stored in well-ventilated areas, kept away from strong oxidizers, and handled with appropriate personal protective equipment.
Comment 19:
Could alternative green reductants (e.g., plant extracts, sugars) substitute VC while maintaining efficiency?
Response 19:
Thank you for your suggestion. Regarding whether alternative green reductants can substitute VC while maintaining efficiency, we evaluated plant extracts and sugars as potential green substitutes for VC in our Cu2O-based system.
Plant extracts: The major advantages of plant extracts are biodegradable and abundant in antioxidants. However, due to variations in source and extraction method, the reducing performance of plant extracts varies significantly, and the cost of purified extracts is relatively high (approximately four times that of VC). Furthermore, the reduction of Cu2+ using plant extracts requires longer reaction times and pH adjustment, which reduces the feasibility of industrial application.
Sugars: Sugars offer the advantages of low cost, non-toxicity, and complete biodegradability. However, their reduction reaction rate is relatively slow, and they are effective only under alkaline conditions, which are not optimal for our acidic wastewater system.
Therefore, in terms of efficiency, cost, and stability, VC remains the most balanced choice. However, we are exploring the extraction of low-cost, high-activity plant extracts from agricultural waste, as well as sugar-based hybrid systems (for instance, glucose-citric acid), to enhance their reactivity under acidic conditions.
Comment 20:
How reproducible are the results across multiple wastewater samples? Did you test seasonal or batch variability in industrial effluents?
Response 20:
Thank you for your suggestion. Regarding the reproducibility of the results, this aspect has been considered in our experimental design. We performed at least three parallel experiments on the same wastewater sample, and the results showed that the Cl⁻ removal efficiency was relatively stable, indicating that the method has good reproducibility. Although we did not systematically investigate seasonal variations, we tested industrial wastewater samples from different sources, including those discharged during different production cycles. The results demonstrated that the Cl⁻ removal efficiency remained above 90%, suggesting that the method has a certain degree of adaptability to common batch-to-batch variations.
The lack of a dedicated assessment of the impact of seasonal changes represents a limitation of this study. In future work, we plan to collect wastewater samples from different seasons for testing, establish an online monitoring system, and develop adaptive control algorithms to address potential water quality fluctuations in practical applications.
Comment 21:
What are the regulatory perspectives does the residual Cu concentration after treatment meet typical discharge standards in China or internationally?
Response 21:
Thank you for your suggestion. The related contents haven been supplemented in lines 391-393 of the manuscript. In this study, the residual Cu2+ concentration after treatment by the sulfide precipitation method was below 1 mg/L, which meets certain typical discharge standards in China and internationally.
In China: According to the Integrated Wastewater Discharge Standard (GB 8978–2002), the maximum allowable discharge concentration of Cu2+ under the third-level standard is 2.0 mg/L. Additionally, China specifies that the maximum allowable discharge concentration of copper and its compounds in industrial wastewater is 1 mg/L (calculated as copper). Furthermore, the Standards for Drinking Water Quality (GB 5749–2022) in China sets a limit of 1 mg/L for copper content. The Cu2+ concentration in this study was below these limits.
Internationally: The World Health Organization (WHO) recommends a copper limit of 2 mg/L for drinking water. For the copper smelting industry, both the European Union and the U.S. Environmental Protection Agency (EPA) have set a copper discharge limit of 2.0 mg/L. The Cu2+ concentration in this study was below these limits.
Reviewer 2 Report
Comments and Suggestions for Authors
Reviewer’s comments
- Line 58, this looks incomplete. What contribute to chloride to be stable and have little biodegradability?
- Line 95 to 98 needs to be referenced.
- Where are figures S1-S4 and Table S1 data, I couldn’t find them on the attached documents. I suggest writing the method on the manuscript as it is a short document.
- Line 120, on the subtitle refrain from using symbols “&”.
- Line 121 why do you wash the material with deionised water.
- Line 146, “An UV-vis spectrophotometer.” is incorrect is a UV-vis. The choice of ‘and an ‘depends on the sound that follows not the letter.
- Line 171, 267, avoid the use of the first person “we” and “our”. Check for similar mistakes throughout the document.
- Fig 1b shows that COA and COB respectively exhibited polyhedral and cubic shapes. What does this mean for the removal of Cl- for each material.
- Include the comparison study on the removal of Cl-
- In your conclusion add the recommendation of the study.
- Avoid the use of old references and include not more than five years references.
Author Response
Comment 1:
Line 58, this looks incomplete. What contribute to chloride to be stable and have little biodegradability?
Response 1:
Thank you for your suggestion. The related contents haven been supplemented in lines 58-62 of the manuscript. As the most stable form of chlorine element, Cl− in acidic aqueous conditions possesses extremely stable electronic structure and high chemical inertness, making its removal relatively difficult. Additionally, for most organisms, Cl− does not participate in energy metabolism, and thus is not utilized by microorganisms as an energy source, nor can it be degraded through metabolic pathways such as respiration or fermentation.
Comment 2:
Line 95 to 98 needs to be referenced.
Response 2:
Thank you for your suggestion. The related references haven been supplemented in lines 522-525 of the manuscript.
Comment 3:
Where are figures S1-S4 and Table S1 data, I couldn’t find them on the attached documents. I suggest writing the method on the manuscript as it is a short document.
Response 3:
Thank you for your suggestion. The related contents haven been supplemented in lines 114-122, lines 127-148, Table 1 and Table 2 of the manuscript.
Comment 4:
Line 120, on the subtitle refrain from using symbols “&”.
Response 4:
Thank you for your suggestion. The related contents haven been modified in lines 149, 311 and 326 of the manuscript.
Comment 5:
Line 121 why do you wash the material with deionised water.
Response 5:
Thank you for your suggestion. The prepared Cu2O was thoroughly washed with DI water to remove residual precursors, byproducts, and adsorbed ions on the surface. This step is crucial to eliminate potential interference from impurities in subsequent experimental processes, ensuring the accuracy and reproducibility of the results. Moreover, washing with DI water avoids the introduction of additional ions that could alter the material’s surface properties or affect its reactivity.
Comment 6:
Line 146, “An UV-vis spectrophotometer.” is incorrect is a UV-vis. The choice of ‘and an ‘depends on the sound that follows not the letter.
Response 6:
Thank you for your suggestion. The related contents haven been modified in line 175 of the manuscript.
Comment 7:
Line 171, 267, avoid the use of the first person “we” and “our”. Check for similar mistakes throughout the document.
Response 7:
Thank you for your suggestion. The related contents haven been modified in the manuscript.
Comment 8:
Fig 1b shows that COA and COB respectively exhibited polyhedral and cubic shapes. What does this mean for the removal of Cl- for each material.
Response 8:
Thank you for your suggestion. COA and COB exhibit different shapes, which implies that the exposed crystal facets may vary. The specific crystal facets exposed by different structures may possess unique surface chemical properties, enhancing the selectivity and affinity for Cl⁻ in acidic wastewater. According to our experimental results, COB exhibits better Cl⁻ removal performance than COA, which may be attributed to the difference in the reactivity of the exposed crystal facets.
Comment 9:
Include the comparison study on the removal of Cl-.
Response 9:
Thank you for your suggestion. The related contents haven been supplemented in lines 383-398 of the manuscript.
Processing cost: According to the preparation procedures of COB, it can be calculated that approximately 3333 g of NaOH, 3472 g of CuSO4·5H2O, and 1222 g of VC are needed to prepare 1 kg of COB, with the reagent cost of approximately 89 rmb. Additionally, for the treatment of 1 ton of wastewater containing 500 mg/L of Cl− and 100 g/L of H2SO4, the reagent cost of this proposed method will be nearly 290 rmb. Furthermore, the generated CuCl can be regarded as a kind of byproduct, which can further lower the treatment cost. In contrast, for the conventional copper slag method, treating 1 ton of wastewater of the same composition would require 1803 g of copper powder and 21127 g of CuSO4·5H2O, with a rather high cost of approximately 421 rmb.
Cl− removal efficiency: The residual Cl− concentration in this study was below 50 mg/L and met the relevant technical specification formulated by China National Resources Recycling Association. However, the Cl− removal efficiency of the traditional copper slag is relatively limited with the residual Cl− concentration always above 150 mg/L. The extremely high residual Cu2+ concentration exceeding 20000 mg/L also poses challenges for the subsequent treatment of wastewater.
Therefore, the proposed method demonstrates clear advantages compared with conventional approaches in terms of Cl− removal efficiency, cost, environmental safety, and ease of operation.
Comment 10:
In your conclusion add the recommendation of the study.
Response 10:
Thank you for your suggestion. The related contents haven been supplemented in lines 445-450 of the manuscript. Before the large-scale application of this proposed method. the relatively poor stability of Cu2O remains a big issue to be solved. Preparing fresh Cu2O prior to wastewater treatment may be a feasible approach. After treatment in this study, the fate of copper includes unreacted Cu2O, CuCl product, and Cu2S3 byproduct. The regeneration and reuse of Cu2O from CuCl and will be addressed in the subsequent studies.
Comment 11:
Avoid the use of old references and include not more than five years references.
Response 11:
Thank you for your suggestion. The related contents haven been modified in References of the manuscript. Most of the literatures published before 2020 has been replaced with the latest research conducted between 2020 and 2025. However, several highly cited papers on reaction mechanisms have been retained to clarify the reaction mechanism of this study.
Round 2
Reviewer 1 Report
Comments and Suggestions for Authors
The authors of the revised article have answered all questions and concerns and made all suggested modifications. Therefore, the manuscript can be accepted for publication.
Reviewer 2 Report
Comments and Suggestions for Authors
The author has addressed all comments as suggested by the reviewer.